# Engineering of Cas12a nuclease variants with enhanced genome-editing specificity

Peng Chen[1◎], Jin Zhou[1◎], Huan Liu[1◎], Erchi Zhou[2◎], Boxiao He[1], Yankang Wu[1], Hongjian Wang[1], Zaiqiao Sun[1], Chonil Paek[1,3], Jun Lei[1], Yongshun Chen[1]*, Xinghua Zhang[2]*, Lei Yin[1]*

1 State Key Laboratory of Virology, Hubei Key Laboratory of Cell Homeostasis, College of Life Sciences, Department of Clinical Oncology, Renmin Hospital of Wuhan University, Wuhan University, Wuhan, China, 2 The Institute for Advanced Studies, College of Life Sciences, State Key Laboratory of Virology, Hubei Key Laboratory of Cell Homeostasis, Wuhan University, Wuhan, China, 3 The Faculty of Life Science, KIM IL SUNG University, Pyongyang, Democratic People's Republic of Korea

◎ These authors contributed equally to this work.
* yongshun2007@163.com (YC); zhxh@whu.edu.cn (XZ); yinlei@whu.edu.cn (LY)

**Data Availability Statement:** The GUIDE-seq raw sequencing reads are available at the Gene Expression Omnibus (GEO) under accession GSE229888

## Abstract

The clustered regularly interspaced short palindromic repeat (CRISPR)-Cas12a system is a powerful tool in gene editing; however, crRNA-DNA mismatches might induce unwanted cleavage events, especially at the distal end of the PAM. To minimize this limitation, we engineered a hyper fidelity AsCas12a variant carrying the mutations S186A/R301A/T315A/Q1014A/K414A (termed HyperFi-As) by modifying amino acid residues interacting with the target DNA and crRNA strand. HyperFi-As retains on-target activities comparable to wild-type AsCas12a (AsCas12aWT) in human cells. We demonstrated that HyperFi-As has dramatically reduced off-target effects in human cells, and HyperFi-As possessed notably a lower tolerance to mismatch at the position of the PAM-distal region compared with the wild type. Further, a modified single-molecule DNA unzipping assay at proper constant force was applied to evaluate the stability and transient stages of the CRISPR/Cas ribonucleoprotein (RNP) complex. Multiple states were sensitively detected during the disassembly of the DNA-Cas12a-crRNA complexes. On off-target DNA substrates, the HyperFi-As-crRNA was harder to maintain the R-loop complex state compared to the AsCas12aWT, which could explain exactly why the HyperFi-As has low off-targeting effects in human cells. Our findings provide a novel version of AsCas12a variant with low off-target effects, especially capable of dealing with the high off-targeting in the distal region from the PAM. An insight into how the AsCas12a variant behaves at off-target sites was also revealed at the single-molecule level and the unzipping assay to evaluate multiple states of CRISPR/Cas RNP complexes might be greatly helpful for a deep understanding of how CRISPR/Cas behaves and how to engineer it in future.

## Introduction

The clustered regularly interspaced short palindromic repeat (CRISPR)-associated protein (Cas) system allows for a wide range of applications for gene modification when guided by RNA and in the presence of protospacer-adjacent motif (PAM) sequence [1–8]. However,

**Funding:** This work was supported by the National Key R&D Program of China (2022YFA1303500 to LY), the National Natural Science Foundation of China (32101196 to PC, 32171210 and 31870728 to LY), the Fundamental Research Funds for the Central Universities (2042022kf1189 to LY), the China Postdoctoral Science Foundation (2021TQ0253 and 2022M712468 to PC, 2022M722473 to JZ). The funders had no role in study design, data collection and analysis, decision to publish, or preparation of the manuscript.

**Competing interests:** The authors have declared that no competing interests exist.

**Abbreviations:** CRISPR, clustered regularly interspaced short palindromic repeat; DSB, double-strand break; dsODN, double-stranded oligodeoxynucleotide; EGFP, enhanced GFP; GUIDE-seq, genome-wide unbiased identification of double-stranded breaks enabled by sequencing; PAM, protospacer-adjacent motif; RFLP, restriction-fragment length polymorphism; RFN, RNA-guided FokI Nuclease; RNP, ibonucleoprotein; WT, wild type.

crRNA-DNA mismatches may also induce cleavage activity of Cas nucleases, resulting in undesired cleavage at off-target sites [9–16]. The off-target effects not only confound the reliability of research experiments in the lab but also have serious implications for clinical utility. So far, several strategies have been developed to improve the specificity of the Cas nucleases, including the end-point selection after the generation of random mutagenesis by the error-prone PCR [17], direct evolution [18,19], engineering of the sgRNAs (ggX20 sgRNAs and tru-sgRNAs) [20,21], delivery of Cas9-sgRNA ribonucleoprotein (RNP) complexes [22,23], usage of dimeric RNA-guided FokI Nucleases (RFNs) [24], and directional screening in human cells [25]. In particular, structure-guided protein engineering is a very useful method to design high-fidelity Cas-nuclease variants, which has been demonstrated in SpCas9, such as SpCas9-HF1 [15], HypaCas9 [26], eSpCas9 [27], HeFSpCas9 [28], SaCas9-HF [29], etc.

In addition to the commonly used type II-A Cas9, the type V-A system Cas12a can also cleave genome DNA efficiently in vivo. Unlike Cas9, Cas12a orthologs (As- (Acidaminococcus sp.) and Lb- (Lachnospiraceae bacterium) Cas12a nucleases) have several distinct features [5–7,9]. First, Cas12a recognizes the thymine-rich PAM sequence upstream of the target region and triggers the cleavage of target DNA in the PAM distal position, generating staggered ends [5]. Second, Cas12a can process its own CRISPR crRNA array into mature crRNAs to mediate gene editing at different genome sites simultaneously and does not require a *trans*-activating crRNA [6,7,30]. Moreover, Cas12a has low off-targeting in human cells due to low crRNA-DNA mismatch tolerance [9,31]. Therefore, Cas12a is considered to be suitable for multiplex gene editing and accurate genome modification.

Although Cas12a nuclease provides a powerful potential for genome engineering, specificity still needs to be improved for clinical application. In the previous studies of genome-wide specificities of CRISPR-AsCas12a nucleases in human cells, several cleavage events happen in the spacer region with a single mismatch base pair [9]. Several AsCas12a variants have been developed to decrease the indels event at the unwanted loci [32–34]. However, mismatches in the PAM-distal region were frequently observed in off-target sites and few strategies were developed to specifically solve the problem. To address the issue, we engineered a hyper fidelity AsCas12a variant with the mutations S186A/R301A/T315A/Q1014A/K414A (termed HyperFi-As) by modifying amino acid residues interacting with the backbone of the target DNA strand and crRNA strand in both the proximal and distal regions to the PAM through structure-guided protein engineering. Using the genome-wide unbiased identification of double-stranded breaks enabled by sequencing (GUIDE-seq) method, we demonstrated that HyperFi-As dramatically reduced the off-targeting in both the proximal and distal regions to the PAM and displayed better fidelity than AsCas12aWT, the recent versions of Cas12a (As ultra, As plus), and the very low off-target Cas9 variant (SuperFi-SpCas9) reported by Taylor etc. in 2022 which attracts great attentions [35]. Furthermore, the results from the new modified methods of single-molecule magnetic tweezers showed that HyperFi-As reduced the binding capacity and decreased the state of R-loop complex at mismatch sites without compromising on-target binding capacity. All these suggested that HyperFi-As could be a very promising accurate CRISPR-Cas genome-editing tool. The strategy that modifying residues in contact with the backbone of the distal end of the crRNA strand combined with mutating non-specific DNA contacts is very useful for increasing Cas12a specificity.

## Results

### Structure-guided protein engineering for high-fidelity AsCas12a

Cas12a has been noted as a very prominent tool for gene editing with low off-targeting. Our previous work identified CeCas12a and other variants with stringent PAM recognition to

further lower the off-targeting [36,37]. However, the relatively high off-targeting still existed in the distal end of the PAM. To further investigate the off-targeting in the proximal and distal end of the PAM, AsCas12a was selected since its high-resolution structure has been solved [38]. Firstly, 4 amino acid residues (S186, R301, T315, and Q1014) with direct hydrogen bonds to the phosphate backbone of the target DNA strand within a 3.0-Å distance at the proximal end of PAM were identified (**S1 Fig**). Among these 4 residues, both R301A and T315A exhibited a tendency for reduced cleavage activity at several mismatched sites compared to AsCas12a WT [39]. To study the effects of these mutations systematically, we constructed 4 single amino acid mutants by alanine substitution (S186A/R301A/T315A/Q1014A) and generated all possible double, triple, and quadruple versions by combining these mutations. Then, we employed a previously reported human cell-based enhanced GFP (EGFP) disruption assay to test their off-targeting [36]. The result showed that all 14 variants retained comparable on-target activities to AsCas12aWT when using the fully matched *EGFP* crRNA (**Fig 1A**). Next,

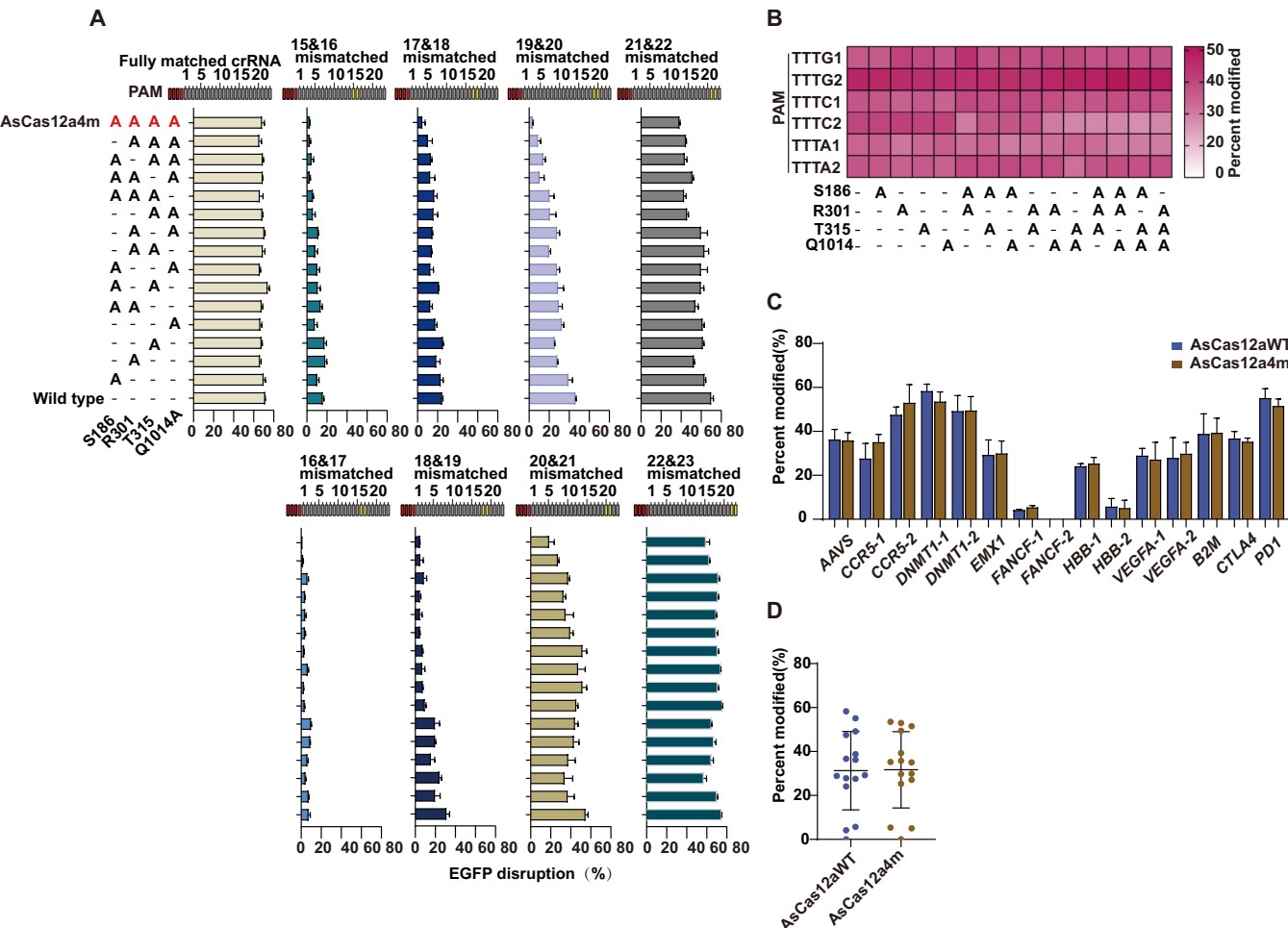

**Fig 1. Identification and characterization of AsCas12a variants bearing substitutions in residues forming nonspecific contacts with target DNA.** (**A**) Characterization of AsCas12a variants containing alanine substitutions in residues that form hydrogen bonds with the target DNA. Assessment of wild-type AsCas12a and variants by using EGFP disruption assay when paired with fully matched crRNA or partially mismatched crRNAs (mean ± SD; $n = 3$). (**B**) Average activity of wild-type AsCas12a and mutant variants on targets with optimal PAM TTTV (V = A; G; C) sequence. (**C**) Activities of wild-type AsCas12a and AsCas12a4m on targets with TTTV PAM sequence across 15 endogenous targets measured by T7 endonuclease I assay (mean ± SD; $n = 3$). (**D**) Summary of on-target modifications from (**C**), with means and 95% confidence intervals shown. The data underlying this figure can be found in S1 Data. EGFP, enhanced GFP; PAM, protospacer-adjacent motif.

we sought to evaluate the performance of these variants toward the imperfect crRNA-DNA matching. To do this, we repeated the EGFP disruption assay by combining all variants with *EGFP*-crRNA expression plasmids that contain 2 substituted bases at positions ranging from 15 to 23 (**Fig 1A**). We found that the nuclease activity of all triply substituted variants reduced dramatically when 2 adjacent crRNA-DNA base pair mismatches were located at the position from 15 to 19 (reduction ranges from 14.1% to 82.5%) (**Fig 1A**). Furthermore, the quadruple substitution variant (S186A/R301A/T315A/Q1014A, termed AsCas12a4m) presented the reduced cleavage efficiency ranging from 20.9% to 92.6% compared to AsCas12aWT (**Fig 1A**).

## AsCas12a4m retains high on-target activities in human cells

Modifying amino acid residues in close contact with the target DNA strand or non-target DNA strand might have various impacts on the cleavage activity at the target site [28]. To systematically determine the activity of AsCas12a4m at endogenous chromosomal sites, we compared the activity of AsCas12aWT and all substitution variants (including AsCas12a4m) at 16 endogenous sites using T7E1 (T7 Endonuclease I) assay in 293T cells (**Figs 1B**, **1C**, **S1** and **S2**). We found all variants retained highly comparable activities (90% to 115%) to AsCas12aWT at TTTV (V = A/G/C) PAM sites (**Figs 1B and S1**), and AsCas12a4m showed at least 90% of the on-target cleavage efficiencies observed with AsCas12aWT at the same sites (**Figs 1C, 1D and S2**). Based on the results, we selected the AsCas12a4m for the subsequent experiment.

## crRNA-DNA mismatch tolerance of AsCas12a4m

For further investigating crRNA-DNA mismatch tolerance of AsCas12a4m in human cells, we arbitrarily selected 2 endogenous target sites (*CFTR* and *B2M*) and constructed a series of plasmids encoding crRNAs with 1 or 2 mismatches (**Fig 2A and 2B**). Firstly, we tested the activity of AsCas12aWT and AsCas12a4m guided by crRNAs with single mismatches along the protospacer complementarity region. For the *CFTR* site, AsCas12aWT showed 5.7% to 68.5% cleavage efficiency at mismatch positions 2, 3, 4, 6, 7, 9, 10, 11, and 13–23 (**Figs 2A and S3**). For the *B2M* site, AsCas12aWT presented 6.9% to 38.1% cleavage efficiency at mismatch positions 1, 8, 9, 10, 11, 13–16, and 18–23 (**Figs 2B and S4**), and did not tolerate single mismatches at positions 2–7 (**Figs 2B and S4**). Not surprisingly, the results demonstrated AsCas12aWT could mediate DNA cleavage with imperfect crRNA-DNA matching at positions 1 through 23, which were consistent with the previous studies [31]. However, for AsCas12a4m, no tolerance was detected for any single mismatches at positions 1–19 (**Fig 2A and 2B**). We also tested the editing of AsCas12aWT and AsCas12a4m at *CFTR* and *B2M* sites by using crRNAs with 2 adjacent mismatches (**Fig 2A and 2B**). The analysis showed that AsCas12aWT and AsCas12a4m processed nearly undetectable cleavage activities with 2 adjacent mismatches through positions 1–18 for both *CFTR* and B2M sites (except for AsCas12aWT at B2M 9 and 10 mismatch) (**Figs 2A, S3 and S4**). By contrast, mismatches (single mismatch or 2 adjacent mismatches) at positions 20–23 did not substantially reduce the cleavage activities for both AsCas12aWT and AsCas12a4m (**Fig 2A and 2B**). Overall, AsCas12a4m exhibited high sensitivity to crRNA-DNA mismatches in both *CFTR* and *B2M* sites especially for the close end region of PAM. The results suggested that AsCas12a4m may enhance the fidelity of gene editing.

## Genome-wide targeting specificity of AsCas12a4m

To evaluate the fidelity of AsCas12a4m at endogenous sites in human cells, we used the GUIDE-seq method to assess the genome-wide specificity of AsCas12aWT and AsCas12a4m with 6 different crRNAs targeted to *PD1*, *B2M*, *HPRT*, *DNMT1*, *CLIC4*, and *NLRC4*. To test

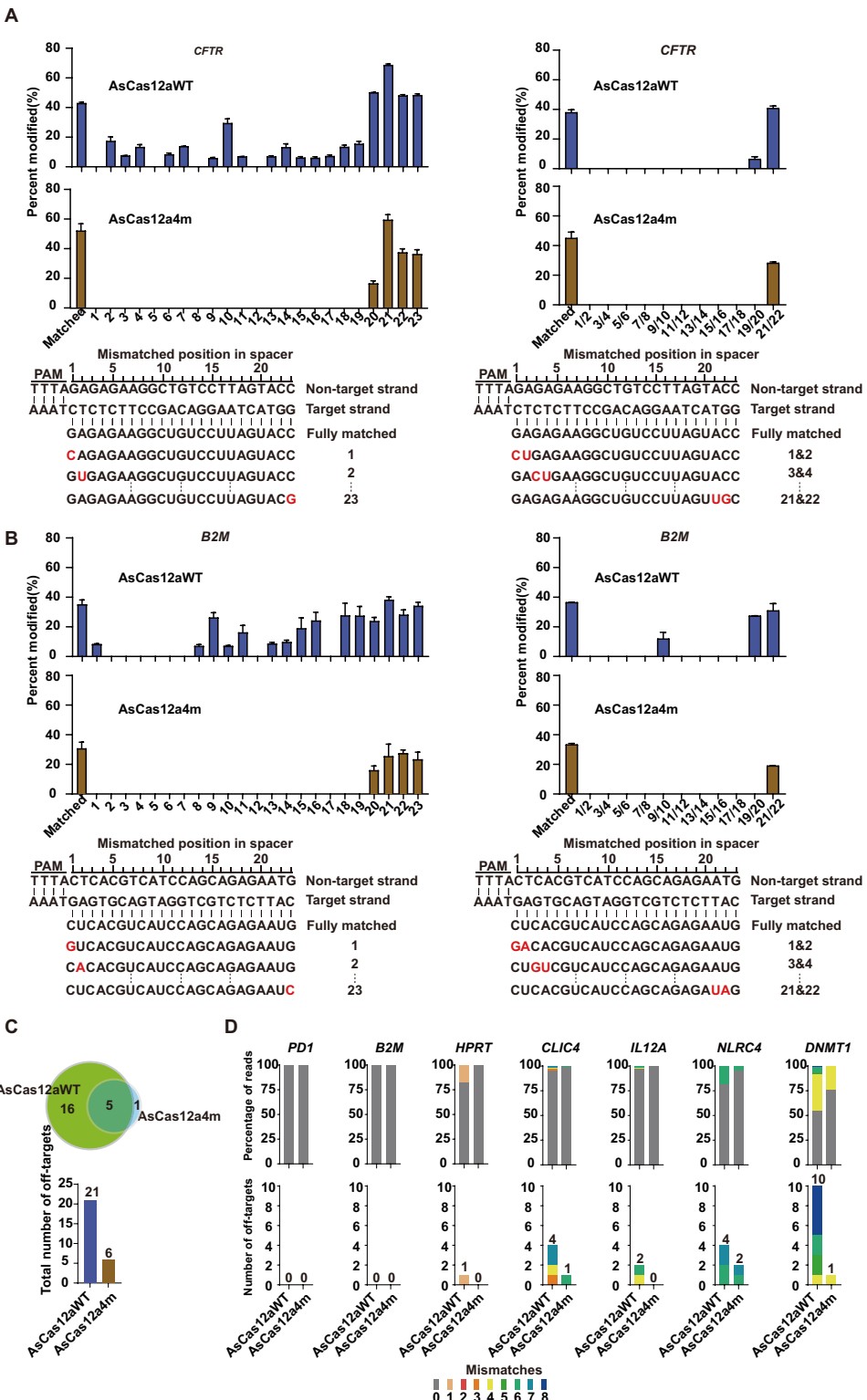

**Fig 2. Tolerance of AsCas12aWT and AsCas12a4m variant to mismatched crRNAs and genome-wide specificities of AsCas12aWT and AsCas12a4m variant with matched crRNAs targeting endogenous sites.** (**A**, **B**) Indels induced by AsCas12aWT and AsCas12a4m using crRNAs that contain singly mismatched bases or pairs of mismatched bases toward *CFTR* (**A**) and *B2M* (**B**). Activity determined by T7 endonuclease I assay. Error bars represent SEM for $n = 3$. (**C**) Summary of the total number of off-target sites identified by GUIDE-seq for AsCas12aWT and AsCas12a4m with

crRNAs targeting 7 endogenous sites. (**D**) Percentage of reads detected by GUIDE-seq at the on-target site and off-target sites (ordered by number of mismatches, 0 represented on-target) among total detected reads by AsCas12aWT, AsCas12a4m (top), and numbers of genome-wide off-target sites (bottom). The data underlying this figure can be found in S1 Data. GUIDE-seq, genome-wide unbiased identification of double-stranded breaks enabled by sequencing.

whether the double-stranded oligodeoxynucleotide (dsODN) tag was integrated into the on-target break site, we performed PCR reactions initiated by 1 primer that specifically annealed to the dsODN and another primer that annealed to the upstream or downstream of on-target break sites (**S5 Fig**). The dsODN tag was successfully integrated into double-strand breaks (DSBs) induced by AsCas12aWT and AsCas12a4m (**S5 Fig**). The GUIDE-seq analysis revealed that crRNAs targeting *PD1* and *B2M* generated no unwanted mutations for AsCas12aWT and AsCas12a4m variant. crRNAs targeting *HPRT*, *DNMT1*, *CLIC4*, and *NLRC4* generated 1, 10, 4, and 4 off-target sites for AsCas12aWT, respectively, while 0, 1, 1, and 2 off-target sites for AsCas12a4m, respectively (**Figs 2C, 2D and S6**). GUIDE-seq analysis of AsCas12a4m showed that the number of off-target sites decreased by lower than 3, the on-to off-target read ratio (range 2% to 25%) was improved, and the number of on-target reads still retained, compared to AsCas12aWT (**Figs 2D and S6**). To validate the off-target sites identified by GUIDE-seq, we confirmed the occurrence of indels using deep sequencing (**S7 Fig**). All results demonstrated that AsCas12a4m possesses higher fidelity than AsCas12aWT.

## Improved specificity of AsRVR and enAsHF1

AsRVR and enAsHF1 have been reported to have a broader targeting range than AsCas12aWT [32,39], while altered PAM preferences might induce extra off-targets at the noncanonical PAM sites [36,40]. We sought to graft AsCas12a4m mutations onto AsRVR and enAsHF1 (AsRVR-4m, enAsHF1-4m) and tested whether these alterations would improve targeting specificity. First, we evaluated the editing activities of AsRVR-4m and enAsHF1-4m in endogenous human genes, and 15 endogenous genomic sites harboring TTTV, TTCV, TATG, TATC, GTTA, CTTA, and GTCA PAMs were selected for T7E1 assay (**Figs 3A, 3B and S8**). As a result, these mutations did not affect the cleavage activity at all of the tested sites (**Fig 3A**). Next, we performed crRNA-DNA tolerance assay, according to the previous method (**Fig 3C and 3D**). Indeed, AsRVR and enAsHF1 exhibited nuclease activity toward crRNA-DNA single mismatch (**Figs 3C, 3D and S9**), and AsRVR showed more sensitivity to crRNA-DNA mismatch at these sites, compared to enAsHF1 (**Figs 3C, 3D and S9**). By contrast, AsRVR-4m and enAsHF1-4m caused a significant decrease in the single mismatch cleavage, which was consistent with the above results (**Figs 2A, 2B, 3C, 3D and S9**).

To globally assess the editing specificity of AsRVR-4m and enAsHF1-4m, we performed GUIDE-seq analysis on 3 endogenous sites that were well studied previously [36,40]. As a result, AsRVR-4m has a dramatically lower number of off-target sites at HEK293 site 1, *POLQ1*, and *POLQ2* (9, 0, and 18, respectively), compared with AsRVR (47, 12, and 77, respectively). Similarly, the number of off-target sites of enAsHF1-4m at the 3 sites (8, 1, and 22) were lower than those of enAsHF1 (36, 7, and 55, respectively) (**Figs 3E and S10**). On- to off-target read ratios of AsRVR-4m and enAsHF1-4m at HEK293 site 1, *POLQ1*, and *POLQ2* significantly increased (57.3%, 100%, and 19.7% for AsRVR-4m and 29.8%, 90.5%, and 19.1% for enAsHF1), compared with AsRVR and enAsHF1(9.6%, 45.6%, 5.8% and 7.5%, 43.6%, 13.3%, respectively) (**Fig 3E–3I**). We performed deep sequencing to validate the off-targets sites identified by GUIDE-seq (**S11 Fig**). As expected, the 4m variant has significantly improved the specificity and dramatically reduced the off-target index (0.67 and 0.65 for AsRVR and

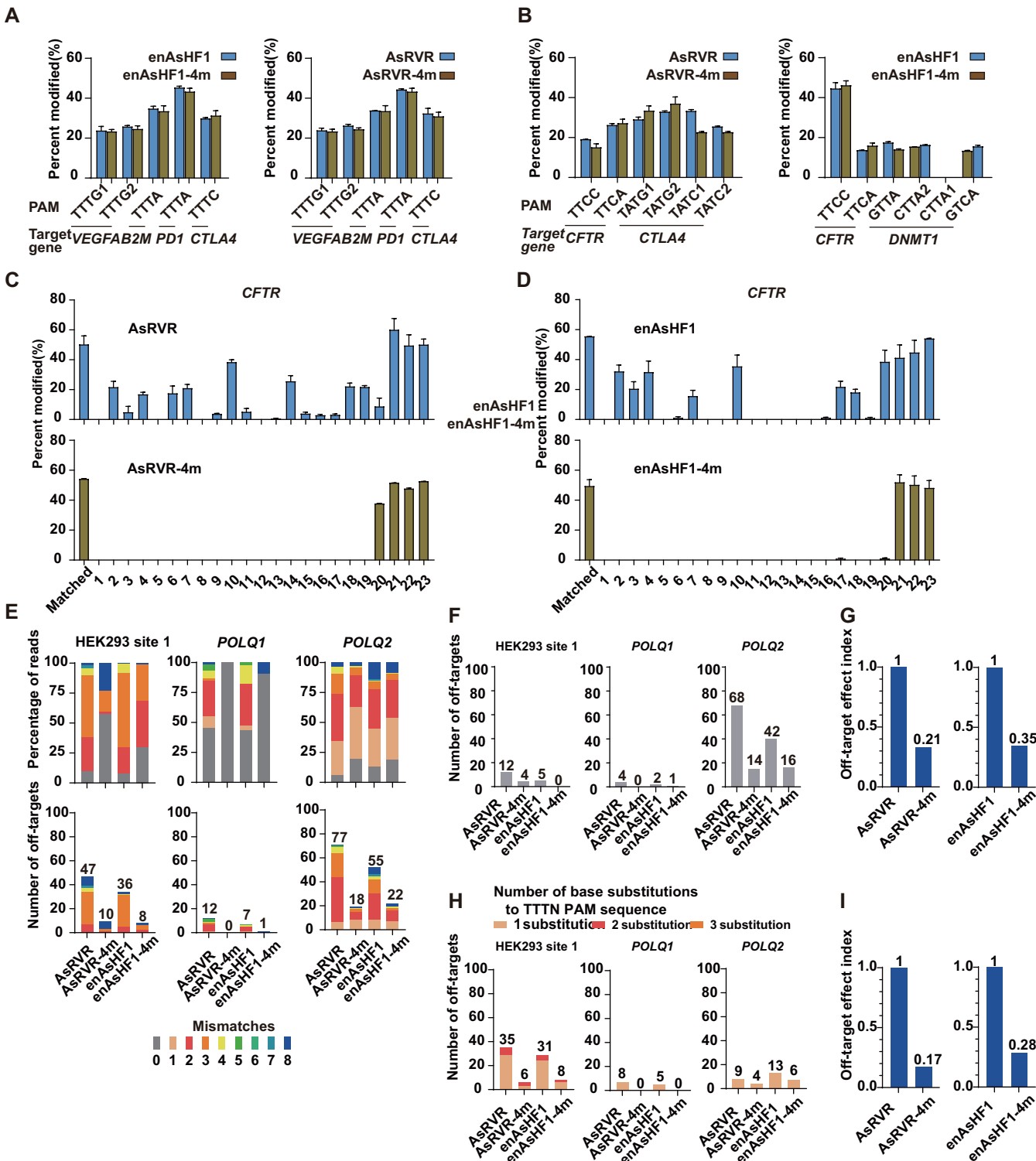

**Fig 3. Refining the specificity of AsRVR and enAsHF1.** (**A**, **B**) Mean on-target percent modification for wild-type AsRVR and AsRVR-4m, enAsHF1 and enAsHF1-4m with crRNAs across sites encoding TTTV PAMs (**A**), TTCV PAMs, TATV PAMs, GTTA PAM, CTTA PAM, GTCA PAM (**B**). (**C**, **D**) Indels induced by AsRVR and AsRVR-4m (**C**), enAsHF1 and enAsHF1-4m (**D**) using crRNAs that contain singly mismatched bases toward *CFTR*. Activity determined by T7 endonuclease I assay. Error bars represent SEM for $n = 3$. (**E**) Percentage of reads detected by GUIDE-seq at the on-target site and off-target sites (ordered by number of mismatches, 0 represented on-target) among total detected reads by AsRVR and AsRVR-4m, enAsHF1 and enAsHF1-4m (top) and numbers of genome-wide off-target sites (bottom). (**F**, **G**) Summary of the total number of off-target sites with TTTN PAMs identified by GUIDE-seq for

AsRVR and AsRVR-4m, enAsHF1 and enAsHF1-4m (**F**) and normalization of off-target effect at TTTN PAMs (value was calculated by the ratio of total off-target sites for AsRVR-4m, enAsHF1-4m to the total off-target sites for AsRVR, enAsHF1 within the detected sites) (**G**). (**H, I**) Summary of the total number of off-target sites with base substitutions to the WT-related TTTN PAMs identified by GUIDE-seq for AsRVR and AsRVR-4m, enAsHF1 and enAsHF1-4m (**H**) and normalization of off-target effect at these sites (**I**). The data underlying this figure can be found in S1 Data. GUIDE-seq, genome-wide unbiased identification of double-stranded breaks enabled by sequencing; PAM, protospacer-adjacent motif.

enAsHF1, for at canonical TTTV PAM, 0.83, and 0.72 for AsRVR and enAsHF1 at noncanonical PAM) (**Fig 3G and 3I**).

## Effects of residues contacting with the 3′ end of crRNA

The cleavage activity of AsCas12aWT nuclease was reported to be nearly unaffected by the mismatches in the PAM distal bases between 18 and 23 [9,31]. We also confirmed it by the T7E1 assay (**Fig 2A and 2B**). To systematically evaluate the effect of the PAM distal bases, we constructed crRNA expression plasmids containing all possible pairs of the mismatched bases between 14 and 23 within the complementarity regions of crRNAs targeted *EGFP* and repeated EGFP disruption assay. In line with our former experiments (**Fig 2A and 2B**), the AsCas12a4m variant showed the high sensitivity to mismatched crRNA nucleotides in most places (**Fig 4A**). However, both AsCas12aWT and AsCas12a4m could mediate enough EGFP disruption when mismatched at the PAM distal positions (**Fig 4A**). GUIDE-seq analysis for AsCas12aWT and AsCas12a4m were further performed at another well-studied endogenous sites *RPL32P3* in 293T cells (**Figs 4B and S12**). To confirm these GUIDE-seq findings, we used T7E1 assay to assess the frequencies of indel mutations induced by AsCas12aWT and AsCas12a4m at 4 off-target sites (**Figs 4C and S12**). In detail, at the off-target sites 3 and 4, the dsDNA cleavage events were 15% and 17% for AsCas12aWT, respectively, and nearly not detected for AsCas12a4m (**Fig 4C**). However, both AsCas12aWT and AsCas12a4m showed enough cleavage events at the off-target sites 1 and 2 (23%, 31%, and 20%, 16% cleavage efficiency for AsCas12aWT and AsCas12a4m at off-target sites 1 and 2, respectively) (**Fig 4C**). The results indicated that: (1) AsCas12a4m-induced dsDNA breaks with high fidelity; and (2) AsCas12a4m was not sensitive to mismatches positions in the PAM distal regions.

To further improve the fidelity in the PAM distal regions, it was initially hypothesized that the unwanted off-targeting with mismatch positions in the PAM distal regions might be minimized by decreasing nonspecific interactions with the 3′ ends of the guide RNA and the 5′ ends of the target DNA. Based on the crystal structure of AsCas12a in complex with crRNA and target DNA [38], we selected 3 residues (K414, Q286, and W382) and 1 residue (S376) that made direct contacts to 3′ ends of the guide RNA and the 5′ end of the target DNA for alanine substitution (**Figs 5A and S13A–S13C**). Then, the on-target activity of 4 mutants over 6 target sites with TTTV PAM sequence were evaluated. As shown in **Fig 5A**, single residue substitutions, except for the W382A hardly perturbed the editing efficiencies at target sites. To evaluate the performance of all possible mutant versions, we performed the T7E1 assay. On-target activity and 4 off-targets activities of all mutant versions were evaluated at the *RPL32P3* target site in 293T cells. For the on-target site, we observed that the indels caused by AsCas12a4m-K414A, AsCas12a4m-S376A, and AsCas12a4m-Q286A mutant were about 32%, 35%, and 31%, respectively, compared to AsCas12aWT, 35% and AsCas12a4m, 33% (**Figs 5B and S13D**). By contrast, other combinations dramatically decreased the activity (**Figs 5B and S13E**). In addition, deep sequencing analysis showed that AsCas12a4m-K414A keep highly comparable activities (80% to 100%) to AsCas12aWT (**S14 Fig**). Indels events did not happen for AsCas12aWT and all As mutants at the off-targeting sites, except AsCas12aWT at the off-target 3 and 4. Furthermore, AsCas12a4m-K414A (termed HyperFi-As) and AsCas12a4m-

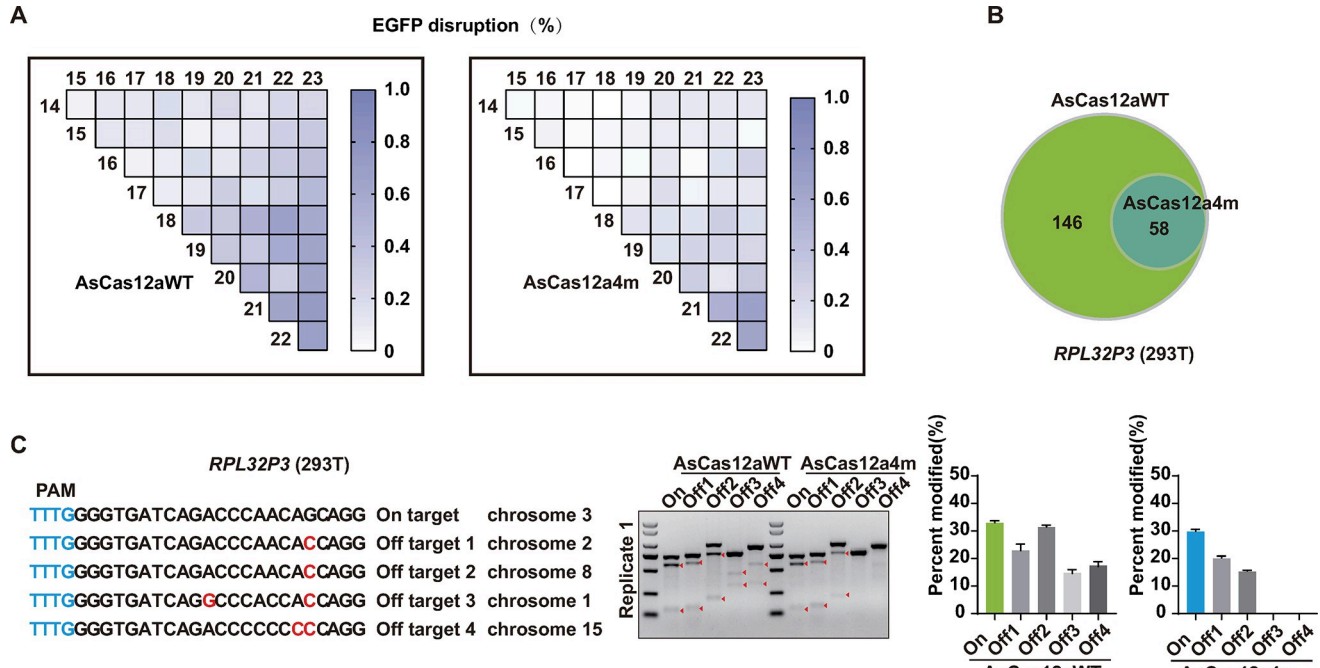

**Fig 4. Performance of AsCas12aWT and AsCas12a4m at PAM-distal mismatches.** (**A**) Cleavage profiles, assessed by EGFP disruption assay for AsCas12aWT and AsCas12a4m at all possible mismatch pairs of bases within the complementarity regions of crRNAs targeted *EGFP* from 14 to 23. The EGFP disruption (%) are the mean of 3 replicates. (**B**) The number of potential off-target loci identified by GUIDE-seq analysis for AsCas12aWT and AsCas12a4m at *RPL32P3* site. (**C**) Validation of AsCas12aWT and AsCas12a4m performance at 4 off-targets that identified by GUIDE-seq in panel (**B**), percent modification of on-target and 4 off-targets of AsCas12aWT and AsCas12a4m assessed by T7E1 assay (from S12 Fig). Error bars represent SEM for *n* = 3. The data underlying this figure can be found in S1 Data. EGFP, enhanced GFP; GUIDE-seq, genome-wide unbiased identification of double-stranded breaks enabled by sequencing; PAM, protospacer-adjacent motif.

Q286A were revealed to have the lower off-target rates at the off target 1 and 2 (**Figs 5B and S13D**). Although further proof of GUIDE-seq analysis indicated that HyperFi-As increased the off- to on-target read ratio at off-target 1, the number of total off-target sites decreased by 13 compared to AsCas12a4m-Q286A (**Figs 5C, 5D, S12C, and S12D**). We further selected *SIPRa*, a well-studied target site to perform GUIDE-seq analysis for specificity assessment [41]. The result demonstrated that targeting *SIPRa* showed off-target activity for HyperFi-As at 3 loci, which was less in number than 49, 9, and 7 for AsCas12aWT, AsCas12a4m, and AsCas12a4m-Q286A, respectively (**Fig 5E and 5F**). In addition, statistics of the GUIDE-seq reads revealed that HyperFi-As possessed the lowest percentage of off-targets with mismatches in the PAM distal region among As, AsCas12a4m, HyperFi-As, and AsCas12a4m-Q286A (**Fig 5F**).

## Systematical comparison of AsCas12a variants and SpCas9 variants

To systematically compare the specificity of HyperFi-As with other recent high-fidelity AsCas12a variants and SpCas9 variants, we comprehensively assessed these variants' capability to reduce genome-wide off-target effects of gRNAs designed against target sites with a large number of homopolymeric sequences in human cells. For the *PPP1R13L* site, HyperFi-As exhibited a reduced number of off-target sites compared to As variants, and the percentage of on-target was also the highest among variants (**Figs 5G and S15**). To further confirm it and compare the genome-wide specificities of CRISPR-Cas nucleases including AsCas12aWT, As ultra, As plus, SpCas9, and SuperFi-SpCas9, we repeated GUIDE-seq with 5 gRNAs targeting

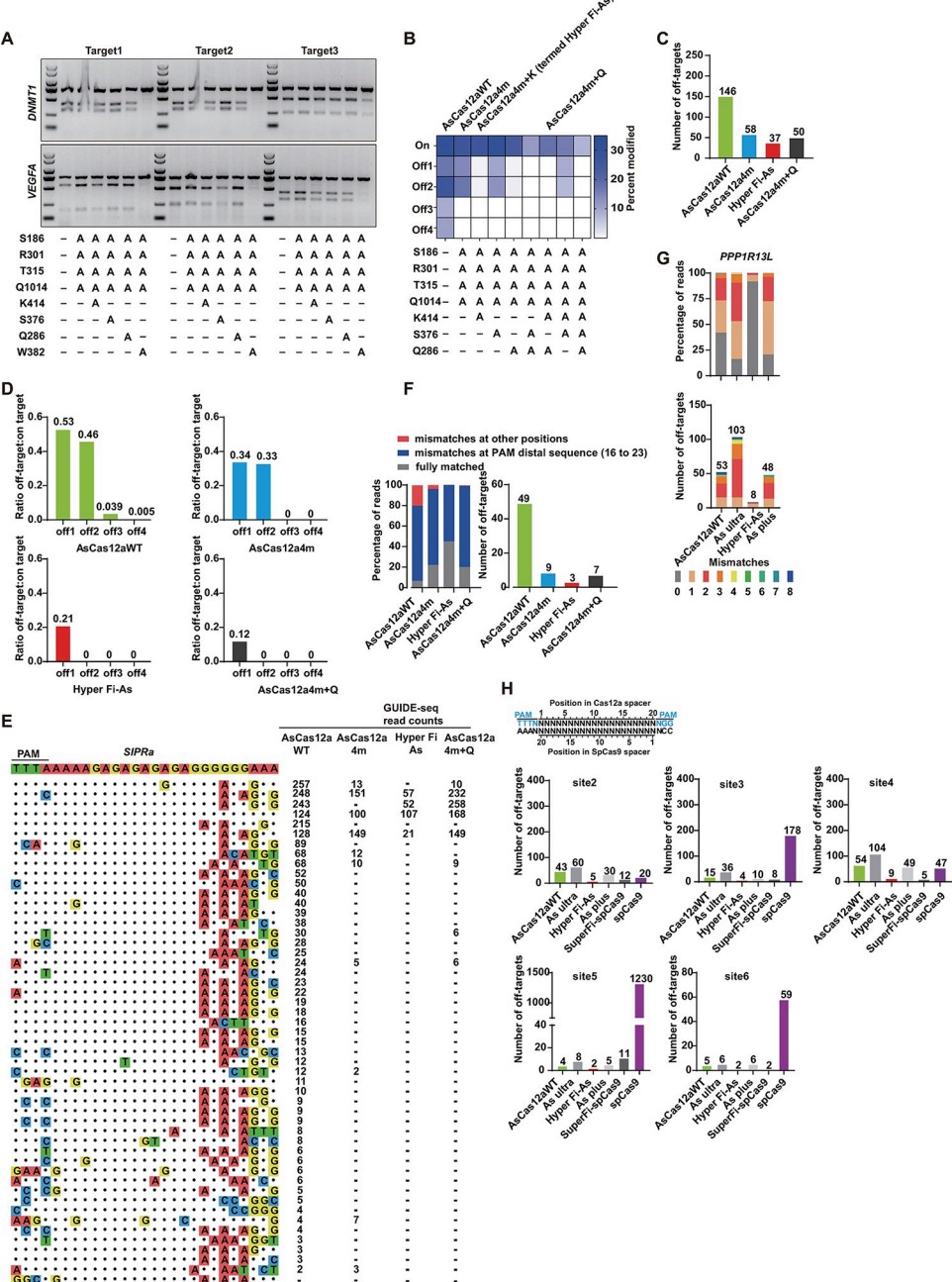

**Fig 5. Generation of high-fidelity derivatives of AsCas12a4m by weakening nonspecific crRNA contacts.** (**A**) Characterization of high-fidelity derivatives of AsCas12a4m containing alanine substitutions in residues that form hydrogen bonds with the PAM-distal region (3′ end of the guide RNA and 5′ end of the target DNA). Assessment of all variants by using T7E1 assay when paired with 6 crRNAs that targeted *DNMT1* and *VEGFA* loci. (**B**) Cleavage profiles, assessed by T7E1 assay for As variants at on-target and 4 off-targets. The percent modification (%) are the mean of 3 replicates. (**C**) The number of potential off-target loci identified by GUIDE-seq analysis for As variants at *RPL32P3* site. The statistical results for AsCas12aWT and AsCas12a4m are derived from Fig 4B (see S12 Fig). (**D**) Specificity ratios of AsCas12aWT, AsCas12a4m, HyperFi-As, and AsCas12a4m+Q, plotted as the ratio of 4 off-target sites GUIDE-seq read counts to on-target. (**E**) Comparative analysis of AsCas12aWT, AsCas12a4m, HyperFi-As, and AsCas12a4m+Q with crRNA targeting *SIPRa* loci using GUIDE-seq. (**F**) Percentage of edited reads detected by GUIDE-seq at on-target site and mismatched sites among total edited reads and the numbers of off-target sites for each AsCas12a. (**G**) Comparative analysis of AsCas12aWT, HyperFi-As, As ultra, and As plus with crRNA targeting *PPP1R13L* loci using GUIDE-seq, percentage of reads detected by GUIDE-seq at the on-target site and off-target sites (ordered by number of mismatches, 0 represented on-target) among total detected reads by As variants (top) and

numbers of genome-wide off-target sites (bottom). (**H**) Genome-wide specificities of AsCas12a and SpCas9 variants. Summary of the total number of genome-wide off-target cleavage sites identified by GUIDE-seq for AsCas12a and SpCas9 variants with gRNAs targeted to nonstandard, repetitive sites. The data underlying this figure can be found in S1 Data. GUIDE-seq, genome-wide unbiased identification of double-stranded breaks enabled by sequencing; PAM, protospacer-adjacent motif.

sites containing overlapping sequences (HEK293 site 2/3/4/5/6) (**Figs 5H and S16–S20**). For 5 sites, the high-fidelity variants such as As plus and SuperFi-SpCas9 exhibited improved specificity (**Fig 5H**), which was consistent with previous reports [33,35,42,43]. However, no on-target gene editing events were detected at the HEK293 site 2/4/6 for SuperFi-SpCas9 (**S16, S18 and S20 Figs**). Notably, HyperFi-As showed the best specificity among the 6 tested variants of AsCas12a and SpCas9 (**Figs 5H and S15–S20**).

## Binding stability and multiple state characteristics of HyperFi-As

To explore whether HyperFi-As has better specificity to distinguish the mutant DNA sequence and thus possibly decreases the off-target efficiency, we compared the stability of DNA-Cas12a-crRNA complexes that containing the HyperFi-As and AsCas12aWT on various mutant DNA sequences using single-molecule DNA unzipping experiments, according to the previous study for determining the stability of DNA-Cas12a-crRNA [44]. However, a proper constant force was used instead for sensitive detection. In the end, the life-time of the intermediate states during the disassembly sensitively detects multiple states by this modified new way.

We constructed a series of 34-bp DNA hairpins containing the PAM and various target/off-target sequences (**Figs 6A and S21A**) as the substrates. We stretched the hairpin between a paramagnetic magnetic bead and a glass slide through 2 dsDNA handles. In the force-increasing scan at the loading rate of 1 pN/s, the naked DNA hairpin unzipped at approximately 18 pN, while a high force of approximately 30 pN was required to completely unzip the DNA-Cas12a-crRNA complex (**S21B Fig**). Note that we used the dead mutants of AsCas12a without cleavage activity and the DNA hairpins were protected.

To determine the stability and possible multiple states of each kind of DNA-Cas12a-crRNA complex, we elevated the force from 5 pN to 26 pN and held the force at 26 pN for 120 s (**Fig 6B**). The measurements were repeated at least 20 times for each molecule, and the results from multiple molecules were yielded. The transient R-loop complex state as the first of 3 states was sensitively detected and 2 steps was illustrated to completely unzip the hairpin that bound crRNA-Cas12a (**Fig 6B**). Based on the increases in DNA extension, the first step represents the dissociation of the non-target sequence from the R-loop complex, and the second step represents the unzipping of PAM (**Fig 6C**). The difference between states 1, 2, and 3 is the length of the non-target sequence bound to Cas12a. In state 1, the Cas12a-crRNA-DNA complex forms a tight binding, encompassing both the target sequence and the non-target sequence. State 2 represents a partial dissociation of the Cas12a-crRNA-DNA complex, with the non-target strand upstream of the PAM being released from the R-loop complex. In state 3, the hairpin downstream of the PAM, including the PAM sequence, unfolds under force and is released from the Cas12a-crRNA complex. The transition from state 1 to state 2 indicates the non-target strand upstream of the PAM moving away from the R-loop complex while the transition from state 2 to state 3 indicates the unfolding of the hairpin strand downstream of the PAM including the PAM sequence. The stabilities of the complexes that had the same crRNA but with 3 different DNA sequences (the WT target and 2 real off-target DNA subtracts) were compared. When bound to the on-target, the instability of DNA-Cas12a-crRNA were similar for both HyperFi-As and AsCas12aWT. However, when bound to the off-target, the instability

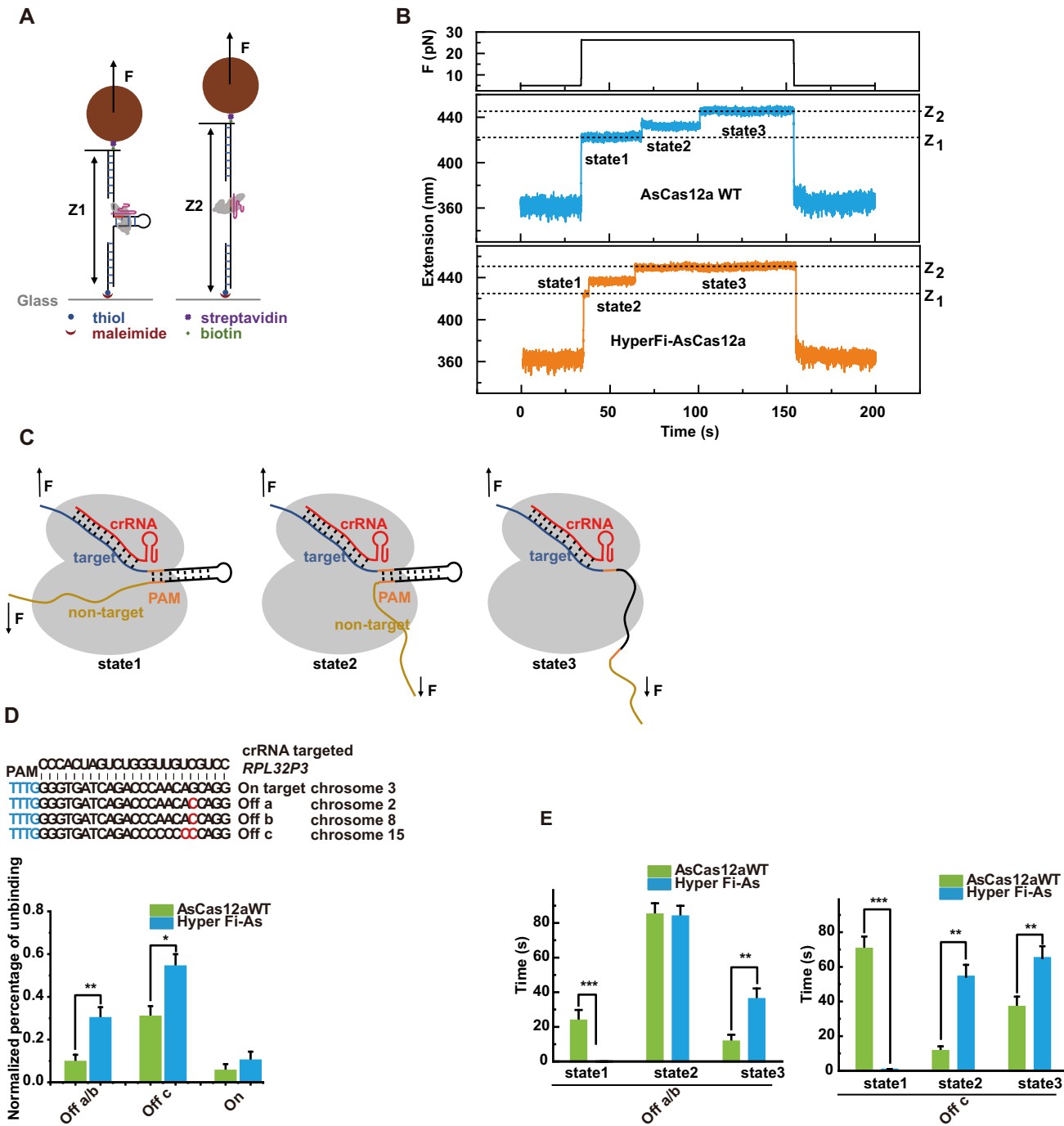

**Fig 6. Single-molecule assay for detection of cas12a complex stability.** (**A**) Sketch of the representation of the experimental setup. A DNA hairpin containing target sequence was attached between the functionalized coverslips and a magnetic bead. As the disassociation of crRNA-Cas12a complex from the target DNA, the extension of DNA molecular stretches with the complete unzip of the hairpin. (**B**) Representative extension trace from the DNA hairpin in the presence of the dCas12a–crRNA complex. States 1, 2, 3 mean the different extension state of the DNA hairpin. (**C**) Cartoon of Cas12a complex in different states. (**D**) Histogram of normalized unbinding events probability on truly mismatch DNA target. (**E**) Histogram of times in different states on truly mismatch DNA target. The data underlying this figure can be found in S1 Data.

was more affected and increased a lot for HyperFi-As (Fig 6D), which was consistent with the result that HyperFi-As had the similar gene editing efficiency at on-target sites but a lower tolerance for off-target sites than AsCas12aWT. Detailed analysis of the unzipping assay revealed that HyperFi-As was largely less distributed at the first state when bound to the off-target sites (Fig 6E), which might suggest more dissociations of the non-target sequence from the R-loop complex for HyperFi-As. All these shed important insights on how HyperFi-As could behave in the single-molecule level and lead to low off-targeting.

## Discussion

Off-target effects of CRISPR-Cas gene-editing tools pose a challenge for therapeutic applications. Although AsCas12a has been shown to be an innately highly specific enzyme, it still causes off-target effects in mammalian cells [31,33,36]. Off-target effects can occur at levels 10 or even 100 times lower than the target site, but these small DNA breaks can cause chromosome breaks and thus chromosomal translocations [37]. Thus, the less off-target we can get, the more safety we can have in terms of therapies. Genome-wide unbiased identification of DSBs by sequencing (GUIDE-seq) is one of the most popular methods to detect real cleavage sites in living cells after genome editing [13]. It is straightforward, has good sensitivity, and reflects the real situations in living cells across the entire genome unbiased. It is therefore well accepted and widely used in many off-targeting studies for Cas9 and Cas12a systems [45–47]. Using GUIDE-seq method, we and other researchers have shown that gene editing in mammalian cells using the Cas12a system causes unwanted mutations, although less so compared to the Cas9 system [9,11,12,31,36,48,49]. Using an in vivo *EGFP* disruption assay, we further clarified the off-target effect of AsCas12aWT in the presence of double base pair mismatches. As mentioned above, we observed significant fluorescence reduction when AsCas12aWT was guided by certain mismatched crRNA, particularly when mismatches were located at the distal end of the PAM sequence. To minimize these off-target effects of AsCas12aWT, we engineered a novel AsCas12a variant termed HyperFi-As that shows high gene editing accuracy without compromising on-target efficiency.

A strategy of decreasing nonspecific interactions with its target DNA strand has been successful for high-fidelity Cas9 [15,29]. Since Cas12a is relatively high in fidelity, it was unclear if the strategy could work well for Cas12a in further improving the fidelity. Thus 4 key residues (S186, R301, T315, and Q1014) was identified by analyzing the crystal structure of the AsCas12a-crRNA-target DNA complex. All of them made direct hydrogen bonds to the phosphate backbone of the target DNA strand within a 3.0-Å distance to provide nonspecific contacts. We constructed single amino acid mutant by alanine substitution (S186A/R301A/T315A/Q1014A) and generated a range of possible combinations and demonstrated that the quadruple substitution variant (termed AsCas12a4m) has the lowest mismatch tolerance by both EGFP disruption assay as well as single mismatch and double mismatch cleavage experiments on endogenous genes. As expected, AsCas12a4m showed a decrease in cleavage activity compared to AsCas12aWT at off-target sites, both in terms of the number of off-target sites and in terms of read counts at the sites, as validated by GUIDE-seq analysis across several endogenous loci.

Recent studies revealed that the alteration of contacts between PAM proximal DNA and amino acid residues in the PI domain of AsCas12a can expand the range of PAM recognition [32,39]. For instance, AsCas12a variants AsCas12a-RVR and enAsCas12a-HF1 modify DNA efficiently at TATV, and TTYN (TTTN/TTCN), VTTV (ATTV/CTTV/GTTV), TRTV (TATV/TGTV) noncanonical PAMs, respectively. However, our previous report indicated that the noncanonical PAM recognition by Cas12a might induce extra off-target edits [36,37].

And the side effect was also demonstrated by other studies [36,40]. In this work, we successfully grafted mutations from AsCas12a4m onto AsRVR and enAsCas12a-HF1 and engineered alternative high-fidelity AsRVR-4m and enAsHF1-4m, which preserved the recognition of nonclassical PAMs.

EGFP disruption assay and cleavage experiments on endogenous sites suggested that one-base mismatches or two-base mismatches at the position of the PAM-distal region were highly tolerant. And truncation of the 3′ ends of the guide RNA could not decrease the undesired mutagenesis at some mismatch sites or at least, so the approach limited, as most off-target sites harbor mismatches in the PAM-distal 3′ end [9,50]. Although AsCas12a4m exhibits low tolerance for mismatches between crRNA and target DNA in other regions, it remains insensitive to mismatches located at the termed promiscuous region (positions 19–23). This observation can be explained by the fact that S186 and Q1014/T315/ R301 form hydrogen bonds with the DNA phosphate backbone of the seed region (positions 1–6) and trunk region (positions 7–18), respectively. We speculated that disruption of nonspecific interactions in the promiscuous regions may also contribute to fidelity. Based on the hypothesis, we finally identified that K414A mutation led to an enhancement of specificity in the PAM-distal end and conferred a higher fidelity to AsCas12a4m by altering the contact with the distal 3′ end of crRNA (termed HyperFi-As). In addition, through the novel single-molecule DNA unzipping experiments demonstrating the binding of DNA-Cas12a-crRNA complex, HyperFi-As mutant significantly reduced the stability of the complexes and decreased the R-loop state when containing the off-target DNA substrate. The new modified single-molecule unzipping assay capable of detecting different state distributions on the disassembly of DNA-Cas12a-crRNA complexes could be widely used in future various studies and could be very helpful for deep insights of how CRISPR/Cas stepwisely heaved and guiding various engineering.

In general, the longer the Cas enzyme and crRNA complex are present in the cell, the greater the likelihood of off-targeting. With virus or plasmid-based delivery, the cellular genome is exposed to the Cas enzyme and crRNA complex for an extended period, which results in an increased risk of off-targeting [51]. Genome editing using Cas12a with RNP or mRNA delivery has been successfully applied to limit the duration of exposure of CRISPR/Cas systems in cells [52,53]. Therefore, introducing the HyperFi-AsCas12a system into cells via RNP or mRNA may further reduce off-target cleavage events.

## Materials and methods

### Plasmids construction

All plasmids and guide RNAs used in this study can be found in **S1 Table**. AsCas12a human expression plasmid pY010 was purchased from the nonprofit plasmid repository Addgene (Addgene plasmids #69982). All AsCas12a variants were generated by standard site-directed mutagenesis. In brief, using AsCas12a plasmid pY010 as the template and a pair of primers that one primer carrying site mutation nucleotide to amplify 500 bp fragments by PCR (Phanta MAX Super-Fidelity DNA Polymerase P505, Vazyme). After confirming that the bands were correct by agarose gel electrophoresis, the PCR products were purified. Next, taking 1,000 ng PCR products as the circling mutation primers and 100 ng AsCas12a plasmid pY010 as template to carry out PCR again. PCR products were purified and use DpnI to digest the original template at 37˚C for 1 h, inactivated at 80˚C for 20 min, take 10 μl digested products for transformation. The next day, single colonies were sent for sequencing. Oligonucleotide duplexes corresponding to spacer sequences were PCR amplified and cloned into pU6-As-crRNA plasmids (Addgene plasmids #78956) for U6 promoter-driven U6 promoter-driven transcription of As crRNAs (ClonExpress II One Step Cloning Kit C112, Vazyme).

## Cell culture and transfection

The HEK293T cells maintained in DMEM medium supplemented with 10% fetal bovine serum and 100 units mL$^{-1}$ penicillin, 100 μg mL$^{-1}$ streptomycin sulfate (all cell culture products were obtained from Gibco) at 37˚C in 5% $CO_2$. For plasmid-based genome editing, approximately $1.2 \times 10^5$ cells were seeded into per well of 24-well plate a day before transfection, 600 ng Cas12a and 300 ng crRNA expression plasmids were transfected into cells using Hieff Trans Liposomal Transfection Reagent (CAT:40802ES03, Yeasen, Shanghai).

## EGFP disruption

For EGFP disruption analysis in HEK293T cells, approximately $2.4 \times 10^4$ cells were seeded into per well of a 96-well plate a day before transfection, a total of 100 ng of Cas12a plasmid, 30 ng crRNA expression plasmids, and 30 ng EGFP expression plasmids were transfected into HEK293T cells. A U6 promoter-driven empty plasmid for the substitution of crRNA expression plasmid as a negative control, and 48 h post-transfection, cells were analyzed on the Cyto-FLEX (Beckman Coulter).

## Assessment of gene editing by T7E1

Approximately 48 h post-transfection, cells were collected by centrifugation and the supernatants were removed. The 50 μl Lysis buffer and 0.5 μl Proteases were mixed with cells and incubated at 55˚C for 30 min, 95˚C for 30 min (Animal Tissue Lysis Component, CAT: 19698ES70, Yeasen, Shanghai). The genomic region flanking the CRISPR target site for each gene was amplified by PCR with Phanta MAX Super-Fidelity DNA Polymerase P505 (Vazyme) using 1 μl cell lysis as template and the primers listed in **S1 Table**. PCR products were purified and the concentrations were determined; 250 ng of purified PCR products were mixed with 1 μl 10×T7E1 buffer (Vazyme) and ultrapure water to a final volume of 10 μl, and subjected to a re-annealing process to enable heteroduplex formation: 95˚C for 3 min, 95˚C for 30 s, 90˚C for 30 s, 85˚C for 30 s, 80˚C for 30 s, 75˚C for 30 s, 70˚C for 30 s, 65˚C for 30 s, 60˚C for 30 s, 55˚C for 30 s, 50˚C for 30 s, 45˚C for 30 s, 40˚C for 30 s, 35˚C for 30 s, 30˚C for 30 s, and 25˚C for 1 min. After re-annealing, products were treated with T7 Endonuclease I (EN303-01/02, Vazyme) for 15 min at 37˚C. The reaction mixtures were run on 2% agarose gels and imaged with ChemiDoc XRS+ and analyzed according to strip intensities. Indel percentage was determined by the formula: $100 \times (1 - sqrt(b + c)/(a + b + c))$, where a is the integrated intensity of the undigested PCR product and b and c are the integrated intensities of the cleavage product.

## GUIDE-seq

Briefly, 600 ng Cas12a, 300 ng crRNA expression plasmids, and 10 pmol of the dsODN GUIDE-seq tag were transfected into a 24-well 293T cells ($1.2 \times 10^5$ cells per well); 48 h after transfection, the genomic DNA was harvested and purified using FastPure Cell/Tissue DNA Isolation Mini Kit DC102 (Vazyme). GUIDE-seq tag integration percentages and on-target modification were assessed by restriction-fragment length polymorphisms (RFLPs) assays and T7E1 assays (as described above), respectively. A total of 1,000 ng genomic DNA was fragmented, end repaired, A-tailing by using Hieff NGS Fast-Pace DNA Fragmentation Reagent (12609ES96, Yeasen, Shanghai). The sequencing library was prepared and sequenced on an Illumina Instrument and data was analyzed using guideseq v1.1 as described previously.

## Western blotting

To detect the expression of As variants, cells were collected and lysed after 48 h transfection. Lysates were resolved through SDS-PAGE electrophoresis and transferred onto a polyvinylidene fluoride membrane (Millipore, United States of America). Membranes were blocked by blocking buffer (5% non-fat milk in Tris buffered saline with Tween20 (TBST)) for 2 h. Blots were incubated with HA Mouse primary antibody (CAT#901515, Biolegend, USA) at 1:20,000 dilution and β-actin Rabbit primary antibody (CAT#AC026, ABclonal, China) at 1:20,000 dilution for 3 h at room temperature, respectively. After washing steps in TBST, membranes were incubated for 1 h with HRP Goat anti-Mouse IgG (H+L) for HA (CAT#AS003, ABclonal, China) and HRP Goat anti-Rabbit for β-actin (CAT#AS014, ABclonal, China) at 1:50,000 dilution, respectively.

## Preparation of DNA samples for MT experiments

The hairpin region contains the PAM and protospacer in the center, schematic of the single-molecule substrate is available in **S21C Fig**.

All oligonucleotides were purchased from Sangon Biotech. The tethered DNA substrate was a molecule comprised of a 34-bp hairpin region, a 5-nucleotide loop, and 2 dsDNA handles, 630 bp and 653 bp, allowing attachment to the glass coverslip and the magnetic bead, respectively. The main step of hairpin construction is showed as follow:

1. We amplified the 653-bp dsDNA3 (6371–7001 bp of λ-DNA) and the 630-bp dsDNA4 (19470–20100 bp of λ-DNA) as hairpin handles through PCR using hairpinF1/hairpin_upR3 and hairpin_downF3/hairpin_R1 as primers, respectively. We then amplify the dsDNA1 containing a part of short hairpin sequences by PCR using template1 (6371–7001 bp of lambda DNA amplified with hairpin_F1 and hairpin_upR1 as primers) and the hairpin_F1, hairpin_upR2 primers. Amplify the dsDNA2 containing the other part of short hairpin sequences by PCR using template2 (19471–20100 bp of lambda DNA amplified with hairpin_downF1 and hairpin_R1 as primers) and the hairpin_downF2, hairpin_R1 primers.

2. The PCR products DNA1 and DNA2 were purified using a Universal DNA Purification Kit (OMEGA Biotech), digested with the restriction enzyme BsaI (NEB), and purified again.

3. Ligation of digested dsDNA1 and dsDNA2 by T4 ligase and form dsDNA5 containing the complete hairpin sequence.

4. Amplify the ssDNA 1 containing the hairpin sequences through OSP using the dsDNA5 as template and hairpin-biotin primers. Amplify the ssDNA 2 through OSP using the dsDNA3 as template and hairpin_upR3 primers. Amplify the ssDNA 3 through OSP using the dsDNA4 as template and hairpin_sh primers.

5. Anneal above 3 ssDNA strands together equimolarly through a process containing a 1-h incubation step at 65°C followed by a 1-h slow cooling process from 65 to 30°C.

   We synthesized all the oligos with the following sequences (Sangon Biotech).
   hairpin-biotin: bio-ATTTACGCCGGGATATGTCAAGC
   hairpin_F1: ATTTACGCCGGGATATGTCAAGCCGAAGCATGAAGTG
   hairpin_upR1:
   TCTGATGGTCCATACCTGTTACACTGCCTGAATGCAGCCATAGGTGC
   hairpin_upR2:
   TGAGGTCTCAAGAAAAACTCACGACGCTTTCTGATGGTCCATACCTGTT

hairpin_upR3: CTGAATGCAGCCATAGGTGC

hairpin-sh: SH-AGTCAGTTGCATCAGTCACAAGGG

hairpin_R1: AGTCAGTTGCATCAGTCACAAGGG

hairpin_downF3: CAGGTATACAGATTAATCCGGC

hairpin_downF1: CATACCTGTTACACTGCCTGAATGCCAGGTATACAGATTAATC CGGC

hairpin_downF2: TGAGGTCTCATTCTCACGACGCTTTCTGATGGTCCATACCTGTT ACACTGCCTGA

## MT experiments

We used a homemade magnetic tweezers setup to stretch individual DNA molecule, which was described previously [54]. We functionalized the coverslips with (3-aminopropyl) triethoxy silane (APTES, Sigma-Aldrich), which allowed the 5′-thiol end of each DNA molecule to attach to the amine group of APTES via Sulfo-SMCC crosslinker (Hunan Huteng). We attached streptavidin-coated paramagnetic beads (Dynal M270, Thermo Fisher Scientific) to the 5′-biotin end of the DNA molecules. We mixed crRNA and Cas12a at 37°C for 20 min in advance and added 10 nM Cas12a-crRNA complexes into the flow cell to form DNA-Cas12a-crRNA complexes. Magnetic tweezers measurements were collected at room temperature (21 to 23°C) in 10 mM Tris-HCl (pH 7.5), 150 mM KCl, and 0.1 mg/ml BSA buffer.

## Statistics analysis and reproducibility

All statistical analyses were performed using GraphPad Prism (v.8.2.1). The exact replication numbers are indicated in the figure legends. The reproducibility was shown by performing at least 2 independent biological replicate experiments.

## Supporting information

**S1 Fig. Identification and characterization of AsCas12a engineered variants.** (**A–E**) Structural representations of AsCas12a-crRNA-DNA complex. In structural representations, amino acid residues (S186, R301, T315, and Q1014) that made direct hydrogen bonds to the phosphate backbone of the target DNA strand within a 3.0-Å distance. Boxes indicate regions shown in detail in **B–E**. Images generated from PDBID:5B43 (ref. [38]) visualized in PyMOL (v 1.8.6.0). (**F**) Activities of AsCas12a engineered variants bearing amino acid substitutions when tested against 7 endogenous sites in human cells. Activities assessed by T7E1 assay. (TIF)

**S2 Fig. Activities of wild-type AsCas12a and AsCas12a4m in human cells.** Comparative analysis of wild-type AsCas12a and AsCas12a variant (AsCas12a4m) with 16 sgRNAs targeting 11 genes. Full gel images of **Fig 1C**. Three independent transfection replicates were done, and activities assessed by T7E1 assay. (TIF)

**S3 Fig. Tolerance of AsCas12aWT and AsCas12a4m variant to mismatched crRNAs targeting *CFTR*.** (**A**) Full gel image for AsCas12aWT activities with single mismatched crRNAs (top) and double mismatched crRNAs (bottom) toward *CFTR*. (**B**) Full gel image for AsCas12a4m activities with single mismatched crRNAs (top) and double mismatched crRNAs (bottom) toward *CFTR*. Three independent transfection replicates were done, and activities assessed by T7E1 assay. Arrows indicates cleavage products. (EPS)

**S4 Fig. Tolerance of AsCas12aWT and AsCas12a4m variant to mismatched crRNAs targeting *B2M*.** (A) Full gel image for AsCas12aWT activities with single mismatched crRNAs (top) and double mismatched crRNAs (bottom) toward *B2M*. (B) Full gel image for AsCas12a4m activities with single mismatched crRNAs (top) and double mismatched crRNAs (bottom) toward *B2M*. Three independent transfection replicates were done, and activities assessed by T7E1 assay. Arrows indicates cleavage products.
(TIF)

**S5 Fig. Examining the integration of GUIDE tag with Tag-specific PCR.** (**A**) Schematic of the Tag-PCR and Genome-PCR. Red arrows indicate genome-specific primers; blue arrows indicate tag-specific primers (Tag-F/R). (**B**) Full gel images of Tag-PCR and Genome-PCR. Red arrow bands indicate the PCR products using the genome-specific primers (B2M-F/R) with tag-specific primers (Tag-F/R). Blue arrow bands indicate the PCR products using the genome-specific primers B2M-F and B2M-R. Con means transfecting without As variants plasmids.
(TIF)

**S6 Fig. Specificity of AsCas12aWT and AsCas12a4m in human cells.** (**A**) Off-target sites for AsCas12aWT and AsCas12a4m with 6 crRNAs targeting 6 endogenous sites (*IL12A*, *B2M*, *PD1*, *CLIC4*, *NLRC4*, *DNMT1*), determined using GUIDE-seq in HEK293 cells. (**B**) Off-target sites for CeCas12a with 4 crRNAs targeting 4 endogenous sites (*B2M*, *PD1*, *CLIC4*, *NLRC4*), determined using GUIDE-seq in HEK293 cells. Mismatched positions are highlighted in color, and GUIDE-seq read counts are shown to the right of the on- or off-target sequences.
(EPS)

**S7 Fig. Deep sequencing validation the off-target sites identified by GUIDE-seq.** Percent modification of on-target and GUIDE-seq detected off-target sites with indel mutations for AsCas12aWT and AsCas12a4m towards *CLIC4* (**A**), *DNMT1* (**B**), and *IL12A* (**C**). Mismatched positions within the spacer or PAM are highlighted in red. Indel frequency assessed by deep sequencing. Error bars represent SEM for $n = 2$. The data underlying this figure can be found in S1 Data.
(EPS)

**S8 Fig. Activities of enAsHF1, AsRVR and enAsHF1-4m, AsRVR-4m in human cells.** (**A**) Activity analysis of enAsHF1, AsRVR and enAsHF1-4m, AsRVR-4m with 5 crRNAs targeting 4 genes (*VEGFA*, *B2M*, *PD1*, *CTLA4*) at TTTV PAMs. Full gel images of **Fig 3A and 3B**, activity analysis of enAsHF1, AsRVR and enAsHF1-4m, AsRVR-4m with 10 crRNAs targeting 3 genes (*CFTR*, *CTLA4*, *DNMT1*) at nonclassical PAMs. Full gel images of **Fig 3B**. Two independent transfection replicates were done, and activities assessed by T7E1 assay.
(TIF)

**S9 Fig. Tolerance of enAsHF1, AsRVR and enAsHF1-4m, AsRVR-4m variants to mismatched crRNAs targeting *CFTR*.** (**A**) Full gel images for enAsHF1 and enAsHF1-4m activities with single mismatched crRNAs toward *CFTR*. (**B**) Full gel images for AsRVR AsRVR-4m activities with single mismatched crRNAs toward *CFTR*. Three independent transfection replicates were done for enAsHF1, AsRVR and enAsHF1-4m and 2 independent transfection replicates were done for AsRVR-4m, activities assessed by T7E1 assay. Arrows indicates cleavage products.
(TIF)

**S10 Fig. Specificity of AsRVR, AsRVR-4m, enAsHF1, enAsHF1-4m, and AsCas12a4m in human cells.** Off-target sites for As variants with crRNAs targeting *POLQ* target 1, *POLQ*

target 2, and HEK293 site 1 loci, determined using GUIDE-seq in HEK293 cells. Mismatched positions are highlighted in color, and GUIDE-seq read counts are shown to the right of the on- or off-target sequences.
(EPS)

**S11 Fig. Deep sequencing validation the off-target sites identified by GUIDE-seq. (A)** On-target cleavages of these variants at 3 endogenous sites evaluated by T7 endonuclease I assay (gel images). (**B**) Percent modification of GUIDE-seq detected off-target sites with indel mutations for enAsHF1, AsRVR, enAsHF1-4m, and AsRVR-4m. Mismatched positions within the spacer or PAM are highlighted in red. Indel frequency assessed by deep sequencing. Error bars represent SEM for $n$ = 2. The data underlying this figure can be found in S1 Data.
(TIF)

**S12 Fig. Specificity of AsCas12aWT, AsCas12a4m, AsCas12a4m+K, AsCas12a4m+Q variants in human cells. (A–D)** Off-target sites for As variants with crRNA targeting *RPL32P3* loci, determined using GUIDE-seq in HEK293 cells. Mismatched positions are highlighted in color, and GUIDE-seq read counts are shown to the right of the on- or off-target sequences. (**E**) Sequences of 4 off-target sites. (**F**) Full gel images for AsCas12aWT and AsCas12a4m activities toward *RPL32P3* on-target site and 4 off-target sites.
(TIF)

**S13 Fig. Identification and characterization of high-fidelity derivatives of AsCas12a4m.** (**A**, **B**) Structural representations of AsCas12a-crRNA-DNA complex. In structural representations, amino acid residues (K414, S376, and Q286) that made direct hydrogen bonds to the phosphate backbone of the PAM-distal region (3′ end of the guide RNA and 5′ end of the target DNA). Boxes indicate regions shown in detail in **B**. Images generated from PDBID:5B43 (ref. [38]) visualized in PyMOL (v 1.8.6.0). (**C**) Expression of As variants in HEK293T cells. (**D**, **E**) Full gel images of AsCas12a4m variants cleavage profiles, 3 independent transfection replicates were done, and assessed by T7E1 assay at on-target and 4 off-targets.
(TIF)

**S14 Fig. Evaluation of the activity of AsCas12aWT, AsCas12a4m, and HyperFi-As. (A)** Validation of HyperFi-As performance at 24 endogenous target sites with TTTV (V = A, C, or G) PAMs in HEK293T cells. (**B**) Schematic of the deep sequencing library constructs. Red arrows indicate genome-specific primers with Illumina P5 and P7 adapter sequences. PCR library is about 280 bp. (**C**) Full gel images of AsCas12aWT, AsCas12a4m, and HyperFi-As deep sequencing libraries. Red box indicated the library band of each endogenous target sites. (**D**) Comparison of the activity of AsCas12aWT, AsCas12a4m, and HyperFi-As in HEK293T cells. Each dot represents a target site. Indel frequency assessed by deep sequencing. The median and interquartile range are shown. (**E**) Evaluation of the activity of AsCas12aWT and HyperFi-AsCas12a with different amounts of plasmid. The data underlying this figure can be found in S1 Data.
(TIF)

**S15 Fig. Specificity of AsCas12aWT, As ultra, As plus, and HyperFi-As in human cells.** Off-target sites for As variants with crRNA targeting *PPP1R13L* loci, determined using GUIDE-seq in HEK293 cells. Mismatched positions are highlighted in color, and GUIDE-seq read counts are shown to the right of the on- or off-target sequences.
(EPS)

**S16 Fig. Specificity comparison of SpCas9 and AsCas12a variants targeting nonstandard, repetitive sites.** Off-target sites for As variants with crRNA targeting HEK293 site 2 loci,

determined using GUIDE-seq in HEK293 cells. Mismatched positions are highlighted in color, and GUIDE-seq read counts are shown to the right of the on- or off-target sequences. (EPS)

**S17 Fig. Specificity comparison of SpCas9 and AsCas12a variants targeting nonstandard, repetitive sites.** Off-target sites for As variants with crRNA targeting HEK293 site 3 loci, determined using GUIDE-seq in HEK293 cells. Mismatched positions are highlighted in color, and GUIDE-seq read counts are shown to the right of the on- or off-target sequences. (EPS)

**S18 Fig. Specificity comparison of SpCas9 and AsCas12a variants targeting nonstandard, repetitive sites.** Off-target sites for As variants with crRNA targeting HEK293 site 4 loci, determined using GUIDE-seq in HEK293 cells. Mismatched positions are highlighted in color, and GUIDE-seq read counts are shown to the right of the on- or off-target sequences. (EPS)

**S19 Fig. Specificity comparison of SpCas9 and AsCas12a variants targeting nonstandard, repetitive sites.** Off-target sites for As variants with crRNA targeting HEK293 site 5 loci, determined using GUIDE-seq in HEK293 cells. Mismatched positions are highlighted in color, and GUIDE-seq read counts are shown to the right of the on- or off-target sequences. (EPS)

**S20 Fig. Specificity comparison of SpCas9 and AsCas12a variants targeting nonstandard, repetitive sites.** Off-target sites for As variants with crRNA targeting HEK293 site 6 loci, determined using GUIDE-seq in HEK293 cells. Mismatched positions are highlighted in color, and GUIDE-seq read counts are shown to the right of the on- or off-target sequences. (EPS)

**S21 Fig. Schematic of the single-molecule assay.** (**A**) The magnetic tweezers setup with the constructed product involving the designed DNA hairpin was tethered between the beads and glass, the DNA hairpin consists a PAM sequence followed by the target sequence of dCas12a-crRNA complex. (**B**) Stretching FECs from the DNA hairpin in the absence (mid) and presence of the dCas12a–crRNA complex (down). (**C**) Schematic representation of hairpin DNA construction for single-molecule experiments. (EPS)

**S1 Data. Individual numerical values of Figs 1–6, S7, S11 and S14.** (XLSX)

**S1 Table. List of sequences used in study.** (DOCX)

**S1 Raw Images. Extended data figures of uncropped gels.** (PDF)

## Acknowledgments

We thank all the members of our laboratory for the fruitful discussions and support.

## Author Contributions

**Conceptualization:** Peng Chen, Jin Zhou, Lei Yin.

**Funding acquisition:** Peng Chen, Jin Zhou.

**Investigation:** Peng Chen, Jin Zhou, Huan Liu, Erchi Zhou, Hongjian Wang, Zaiqiao Sun, Chonil Paek, Jun Lei.

**Methodology:** Peng Chen, Jin Zhou, Huan Liu, Erchi Zhou, Yankang Wu, Hongjian Wang, Zaiqiao Sun, Chonil Paek, Jun Lei, Yongshun Chen, Xinghua Zhang.

**Supervision:** Yongshun Chen, Xinghua Zhang, Lei Yin.

**Validation:** Huan Liu, Erchi Zhou, Boxiao He.

**Visualization:** Erchi Zhou, Yankang Wu.

**Writing – original draft:** Peng Chen, Huan Liu.

**Writing – review & editing:** Peng Chen, Jin Zhou, Lei Yin.

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
