## [Editor Report · Decision Letter 0]

27 Jun 2023

Dear Dr Yin, 

Thank you for submitting your manuscript entitled "Rational designs of Cas12a nuclease with the improved targeting specificity at both the proximal and distal region from the PAM" for consideration as a Research Article by PLOS Biology.

Your manuscript has now been evaluated by the PLOS Biology editorial staff, as well as by an academic editor with relevant expertise, and I am writing to let you know that we would like to send your submission out for external peer review.

Once your full submission is complete, your paper will undergo a series of checks in preparation for peer review. After your manuscript has passed the checks it will be sent out for review. To provide the metadata for your submission, please Login to Editorial Manager (https://www.editorialmanager.com/pbiology) within two working days, i.e. by Jun 29 2023 11:59PM.

Kind regards,

Richard

Richard Hodge, PhD

rhodge@plos.org

PLOS

---

## [Decision Letter · Decision Letter 1]

8 Aug 2023

Dear Dr Yin,

Thank you for your patience while your manuscript "Rational designs of Cas12a nuclease with the improved targeting specificity at both the proximal and distal region from the PAM" was peer-reviewed at PLOS Biology. Please accept my sincere apologies for the delays that you have experienced during the peer review process. Your manuscript has now been evaluated by the PLOS Biology editors, an Academic Editor with relevant expertise, and by three independent reviewers. 

In light of the reviews, which you will find at the end of this email, we would like to invite you to revise the work to thoroughly address the reviewers' reports.

As you will see, the reviewers are generally positive about the findings and think that the high-fidelity AsCas12a variant has the potential to be useful for the field. However, Reviewer #1 raises concerns with overall evaluation of the editing efficiencies of the Cas12a enzyme and asks that deep sequencing is used to analyse editing efficiency and indel frequency at endogenous sites. After discussions with the Academic Editor, we agree that NGS data should be applied for a few very important endogenous targets as outlined by the reviewer to strengthen the confidence in the specificity of the variant, but we will not make a comprehensive evaluation essential to consider a revised version of the manuscript.

Given the extent of revision needed, we cannot make a decision about publication until we have seen the revised manuscript and your response to the reviewers' comments. Your revised manuscript is likely to be sent for further evaluation by all or a subset of the reviewers.

**IMPORTANT - SUBMITTING YOUR REVISION**

*Re-submission Checklist*

*Published Peer Review*

*PLOS Data Policy*

*Blot and Gel Data Policy*

Sincerely,

Richard

Richard Hodge, PhD

rhodge@plos.org

REVIEWS:

Reviewer #1: Chen et al. developed engineered versions of AsCas12a by introducing alanine substitutions to residues involved in the interaction with sugar and phosphate groups of the target DNA and crRNA. The resulting AsCas12a variants, As4m and HyperFi-As, demonstrated enhanced genome editing specificity compared to the wildtype AsCas12a. However, the genome editing efficiencies of As4m and HyperFi-As at on-target sites have not been definitively evaluated. Furthermore, additional modifications are required to clarify the manuscript for a broader readership. Therefore, I strongly recommend that the authors address these points to further improve the manuscript.

1. A comprehensive evaluation of the general activity of As4m and HyperFi-As, as well as the wildtype AsCas12a, is needed. Considering the inherent tradeoff between on-target activity and specificity in most Cas9 variants, it is important to examine this in detail. Evaluating the indel frequency at a minimum of 20-30 endogenous sites would be highly suitable for this purpose. Additionally, I strongly recommend utilizing deep sequencing instead of the T7E1 assay, to enhance the reliability of the findings.

2. In Figure 3D and related figures, quantitatively evaluating genome editing efficiency using the read count from Guide-seq poses challenges. To validate the off-target sites identified by Guide-seq, it is necessary to confirm the occurrence of indels at the endogenous sites using deep sequencing.

3. Figure 5 lacks completeness and contains multiple inaccuracies, making it difficult for readers to understand. The description in the main text does not match the results shown in Fig 5B and Fig 5G. Additionally, Fig 5D is not mentioned anywhere in the manuscript.

4. Additional experiments are required to compare the specificity of SpCas9, AsCas12a, and their derived variants since on-target activity can affect specificity. Although the target site chosen by the authors contained PAM sequences for both SpCas9 and AsCas12a, the spacer sequences are in opposite orientations, resulting in position-dependent differences in nucleotide composition, which may lead to variations in on-target activity. Therefore, it is necessary to investigate whether SpCas9, AsCas12a, and their variants exhibit similar indel frequency values at the on-target sites.

5. Further explanations and revisions are needed for the interpretation of the results in Figure 6. It would be helpful to provide additional explanations about the meaning of state 1, 2, and 3, as well as the transitions from state 1 to 2 and from state 2 to 3. The y-axis in Figure 6D is unclear and hinders understanding of the content. Is it the ratio of RNP molecules that do not form R loop complexes? Furthermore, there is a need to clarify the meaning of terms such as "instability." Lastly, the addition of a figure showing the state transitions of WT AsCas12 and the variants, similar to Fig 6B, would be beneficial.

6. The references need overall organization and revision. (Line 40-41) Reference 27 is not about eSpCas9, and Reference 29 is about SaCas9, so the corresponding sentences need to be modified. (Line 56) Reference 15 cannot provide support for the specificity of AsCas12a. Throughout the manuscript, preceding studies on the structural analysis of AsCas12a are also not cited.

Minor concerns:

(Line 110) "wild-type SpCas9" should be changed to "wild-type AsCas12a."

(Line 160) There is inconsistency in the labeling of the AsRVR and enAsHF1 experimental groups between the manuscript and Figure 3, making it difficult to understand the content.

(Line 214) The manuscript and Figure 5 have different labels for W382 and W342, so correction is needed.

Reviewer #2: CRISPR-Cas12a/Cpf1 enzymes are naturally high specificity nucleases that offer advantages over Cas9-based systems. Although one can argue whether their off-target potential is a problem worth solving (given the generally low levels of off-targets), engineered enzymes that preserve on-target editing while minimizing off-target edits are advantageous for obvious reasons. Here, Chen et al. seek to develop higher-fidelity Cas12a enzymes that have reduced off-target editing compared to the several other versions previously described. This manuscript takes a nearly identical blueprint to that described for SpCas9-HF1 (PMID: 26735016) with similar figures and assays. The result is a highly efficient and specific Cas12a enzyme that has reduced off-target potential. The manuscript is well written, the HyperFi-As enzyme appears to be quite useful. and the data presentations are generally clear; thus, I have few comments. 

Comments:

1. Naturally high fidelity. Some additional commentary from the authors about whether this is a problem worth solving would be helpful. The author's own data shows that in most cases there are either no off-targets, or very few at 10- or 100-fold lower levels than the on-target site. These results are consistent with many prior reports that Cas12a is naturally a high specificity enzyme. This referee is not suggesting that this isn't an important problem, but the authors should more clearly describe why even low level off-targets may be a concern. 

2. On-target activity. The authors's data show that in general HyperFi-As tends to retain on-target editing. Under what conditions were these assays performed (presumably high levels of plasmid delivery into HEK 293 cells)? Some discussion about the likelihood of the enzyme maintaining on-target efficiency in different contexts (e.g. mRNA or RNP delivery) would be helpful. If the authors have data to show that HyperFi-As works well in the mRNA or RNP format, it would be nice to add this data to the manuscript. 

3. Is there a way that GUIDE-seq is missing off-targets for Cas12a or other enzymes that leave staggered breaks?

4. Some of the other improved specificity Cas12a enzymes that have been developed should be discussed earlier in the manuscript (briefly in the introduction). Many of these papers are mentioned later in the results, but since this work is continuing a long line of efforts by other groups, it would be helpful for the authors to add more context. For instance, AsCas12a Plus from Zhili Rong's group (35468792), the HF variant from Joung's group (PMID 30742127), unpublished but public information on AsCpf1 from Joung's group (https://patents.justia.com/patent/20220025347), AsCas12a Ultra (34162850) and others. This prior work should be mentioned in the introduction, given that there is some overlap for the amino acid substitutions selected in this current work. 

5. The R-loop stability data is nice. Does this mean that the HyperFi-As enzyme may not work as well as a base editors? Or that it will be extra specific as a base editor?

6. Citations are missing for many statements in the introduction and throughout the manuscript. For example:

a. Line 47 - PMID: 26422227

b. Line 49 - PMIDs: 26422227, 27918548, 27096362, 29083402, etc.

c. Line 51 - PMIDs: 27272384, 27347757, etc.

d. Etc

Reviewer #3: In this manuscript, Chen et al. report their work on the structure-guided engineering of AsCas12a to enhance the specificity for genome editing in human cells. Through alanine substitution of residues involved in interaction with the target DNA and crRNA at the PAM-distal end, they have made an AsCas12a variant with five point mutations S186A/R301A/T315A/Q1014A/K414A (named HyperFi-As). Their experiments demonstrate that this variant maintains on-target activity while reducing off-target effects.

Overall, the topic of reducing off-target effects is of significant interest in the genome editing field. This study contributes valuable data by analyzing Cas12a and comparing it with various other Cas12a and Cas9 variants aimed at addressing the same challenge. The strength of this research lies in this comparative analysis. However, my primary concern is the extent of improvement offered by this new Cas12a variant relative to previously reported ones. Although the authors claim their Cas12a variant to be superior, the robustness of this variant is unclear. It remains to be seen whether the reduced off-target rate is applicable across other cell lines. While I am not necessarily asking the authors to test additional cell lines in this paper, although such an effort would be creditable, it would be beneficial to situate their findings within a broader context. Discussion of the limitations of their work, like a more quantitative analysis of the observed improvement and the need for testing more cell lines, would be welcome.

In addition, I have the following specific comments for the authors' consideration.

1) Page 3, line 41: The authors provide several examples of structure-guided protein engineering of Cas9. Could they briefly summarize the principles utilized for the engineering? Were the residues involved in binding the PAM-distal end of target DNA and crRNA also mutated?

2) Page 4, line 50: The authors claim that Cas12a exhibits low off-target effects. Is this in comparison to wild type Cas9? They attribute the low off-target activity to 'low crRNA-DNA mismatch tolerance', but do not provide citations. Could they elaborate on the reasoning behind the low off-target effects in Cas12a?

3) Page 8, line 80: When referencing 'our previous work', please provide a citation. The same applies to 'our former experiment' on page 10.

4) Page 6, lines 91-101: Could the authors provide a more quantitative description of the results? When they say 'dramatically', it is unclear to the reader the extent of the difference observed. Consider using more quantitative description for a few other results as well, for example, 'showed enough cleavage events' on page 11.

5) Page 7, line 121: Could the authors explain how they measured cleavage efficiency in human cells? It seems to me that direct cleavage is not directly observable. Instead, the indel rate might be an indirect reflection of cleavage. If my understanding is correct, please revise the text for better accuracy.

6) Page 12, line 221: I recommend using 'indels' instead of 'in/dels', although the decision is up to the authors.

7) Page 13, line 260: There is a typo in 'intermediate states' (not 'intermediated states').

8) Page 14, line 276: As far as I understand, 'the unzipping of PAM' is not a step in Cas12a activation and is therefore confusing here.

9) Page 15, line 295: When indicating that Cas12a unwanted mutation rates are lower than those of Cas9, please provide references.

10) I recommend using 'AcCas12aWT' instead of 'AsWT' and 'AsCas12a4m' instead of 'As4m'.

---

## [Decision Letter · Decision Letter 2]

13 Dec 2023

Dear Dr Yin,

Thank you for your patience while we considered your revised manuscript "Rational designs of Cas12a nuclease with the improved targeting specificity at both the proximal and distal region from the PAM" for publication as a Research Article at PLOS Biology. Please accept my apologies for the delays that you have experienced during this round of the peer review process. This revised version of your manuscript has been evaluated by the PLOS Biology editors, the Academic Editor and the original reviewers.

Based on the reviews, I am pleased to say that we are likely to accept this manuscript for publication, provided you satisfactorily address the remaining points raised by Reviewer #2. This includes ensuring that all of the responses provided in the rebuttal are reflected in the manuscript itself and addressing the requests to provide additional discussions and reporting details. 

In addition, please also make sure to address the following data and other policy-related requests that I have provided below (A-G):

(A) We would like to suggest the following modification to the title: 

““Engineering of Cas12a nuclease variants with enhanced genome editing specificity”

(B) You may be aware of the PLOS Data Policy, which requires that all data be made available without restriction: http://journals.plos.org/plosbiology/s/data-availability. For more information, please also see this editorial: http://dx.doi.org/10.1371/journal.pbio.1001797

-Supplementary files (e.g., excel). Please ensure that all data files are uploaded as 'Supporting Information' and are invariably referred to (in the manuscript, figure legends, and the Description field when uploading your files) using the following format verbatim: S1 Data, S2 Data, etc. Multiple panels of a single or even several figures can be included as multiple sheets in one excel file that is saved using exactly the following convention: S1_Data.xlsx (using an underscore).

-Deposition in a publicly available repository. Please also provide the accession code or a reviewer link so that we may view your data before publication. 

Figure 1B-E, 2A-D, 3A-I, 4A, 4C, 5B-H, 6B, 6D-E, S7A-C, S10A-C, S14D

(C) Please also ensure that each of the relevant figure legends in your manuscript include information on *WHERE THE UNDERLYING DATA CAN BE FOUND*, and ensure your supplemental data file/s has a legend.

(D) We require the original, uncropped and minimally adjusted images supporting all blot and gel results reported in the following Figures:

Figure 4C, 5A, S1F, S2, S3A-B. S4A-B, S5B, S8A-B, S9A-B, S11A, S12F, S13C-E, S14A, S14C

We will require these files before a manuscript can be accepted so please prepare and upload them now. Please carefully read our guidelines for how to prepare and upload this data: https://journals.plos.org/plosbiology/s/figures#loc-blot-and-gel-reporting-requirements

(E) Please ensure that your Data Statement in the submission system accurately describes where your data can be found and is in final format, as it will be published as written there. 

(F) Please note that per journal policy, the species studied should be clearly stated in the abstract of your manuscript (e.g. Acidaminococcus sp).

(G) Please also provide a blurb which (if accepted) will be included in our weekly and monthly Electronic Table of Contents, sent out to readers of PLOS Biology, and may be used to promote your article in social media. The blurb should be about 30-40 words long and is subject to editorial changes. It should, without exaggeration, entice people to read your manuscript. It should not be redundant with the title and should not contain acronyms or abbreviations. For examples, view our author guidelines: https://journals.plos.org/plosbiology/s/revising-your-manuscript#loc-blurb

We expect to receive your revised manuscript within two weeks. 

*Published Peer Review History*

*Press*

Sincerely,

Richard

Richard Hodge, PhD

rhodge@plos.org

Reviewer remarks:

Reviewer #1: In the revised manuscript, the authors have conducted a comprehensive evaluation of the engineered versions of Cas12a nuclease through additional wet experiments and analysis. The implementation of Cas12a variants described in the study would significantly improve the accuracy and safety of genome engineering. I recommend considering this study for publication in PLOS Biology.

Reviewer #2: I feel that the Authors have made inadequate revisions to their manuscript. The authors should not just respond to the Referee comments in their rebuttal letter - they must adjust their manuscript to clarify their text in ways that reflect the requests of the Referees. The Referees have asked various questions for very specific reasons to help improve the clarity of the study - not just for a response in the rebuttal. These points need to be made more clear throughout the manuscript. Please adjust the text accordingly in light of all initial reviewer comments.

Some separate comments:

1. Lines 58/59 - This modification is incorrect. AsCas12a high-fidelity variants have been developed to reduce off-target editing at unwanted loci, not "to decrease the indels even at the spacer region", which implies the on-target site. 

2. One experiment the authors could do to simulate lower expression levels / more transient enzyme exposure (as would be observed by RNP or mRNA delivery) would be to titrate their plasmid delivery down to see whether the engineered enzyme has poorer on-target activity compared to wild-type. In your Discussion, please add some discussion of the translatability of these enzymes into different delivery formats.

3. In your Discussion, add some text to discuss why you think GUIDE-seq is an adequate assay to detect Cas12a off-target sites (and could cite prior work). This is frequently discussed in the genome editing field and should be a point of discussion in this work given how many conclusions are dependent on the results of the GUIDE-seq assay.

4. In the results section when the authors describe which amino acid substitutions they are using in their modified enzyme(s), they should list specific fidelity-enhancing mutations that have been described by prior groups in published literature, so that the readership can best understand what mutations are new compared to what mutations have been described by others.

Reviewer #3: The authors have addressed my previous concerns.

---

## [Editor Report · Decision Letter 3]

22 Jan 2024

Dear Dr Yin,

Thank you for the submission of your revised Research Article "Engineering of Cas12a nuclease variants with enhanced genome editing specificity" for publication in PLOS Biology. On behalf of my colleagues and the Academic Editor, Bon-Kyoung Koo, I am pleased to say that we can in principle accept your manuscript for publication, provided you address any remaining formatting and reporting issues. These will be detailed in an email you should receive within 2-3 business days from our colleagues in the journal operations team; no action is required from you until then. Please note that we will not be able to formally accept your manuscript and schedule it for publication until you have completed any requested changes.

PRESS

Sincerely, 

Christian

Christian Schnell (on behalf of Richard who is out of office currently)

Senior Editor

PLOS Biology

cschnell@plos.org

Richard Hodge, 

Senior Editor

PLOS Biology

rhodge@plos.org